# Unmet need for family planning and associated factors among currently married women of reproductive age in Bishoftu town, Eastern Ethiopia

Megersa Girma Garo[1], Sileshi Garoma Abe[2], Worku Dugasa Girsha[2], Dawit Wolde Daka[3]*

1 Bishoftu Defence Hospital, Bishoftu, Ethiopia, 2 Adama Hospital Medical Science College, Adama, Ethiopia, 3 Faculty of Public Health, Jimma University, Jimma, Ethiopia

* dave86520@gmail.com

## Abstract

### Background

Unmet family planning is one of the common causes for low contraceptive prevalence rates in developing countries, including Ethiopia. Rapid urbanization had profound effect on population health, however, little is known about the unmet need of family planning in settings where there was increased industrializations and internal migrations in Ethiopia. This study aims to determine the unmet need for family planning services among currently married women and identify factors associated with it in Bishoftu town, Eastern Ethiopia.

### Methods

Community-based cross-sectional study was conducted from 1st January to 28th February, 2021 among 847 randomly sampled currently married women of the reproductive age group. Data were collected using semi structured interviewer administered questionnaire. Multivariate logistic regression was used to identify factors associated with the outcome variable and a 95% confidence interval was used to declare the presence of statistical significance associations.

### Results

Eight hundred twenty-eight women were participated in the study. The prevalence of unmet need for family planning among currently married women was 26% [95% CI: 23,29]. Maternal age [AOR, 3.00, 95% CI:1.51–5.95], educational status [AOR, 2.49, 95% CI:1.22–5.07], occupational status of self-employee [AOR, 1.98, 95% CI:1.15–3.39] and housewife [AOR, 1.78, 95% CI:1.02–3.12], being visited by health care provider in the last 12 months [AOR, 1.81, 95% CI: 1.26–2.60] and desired number of children less than two [AOR, 1.53, 95% CI:1.01–2.30] were significantly associated with unmet need for family planning.

**Data Availability Statement:** All relevant data is within the manuscript and supporting information files.

**Funding:** The author(s) received no specific funding for this work.

**Competing interests:** The authors have declared that no competing interests exist.

## Conclusions

Unmet need for family planning was higher in the study area compared with the United Nations sphere standard of unmet need for family planning and the national average, and slightly lower than the regional average. Socio-demographic, economic, and health institution factors were determinants of the unmet need for family planning in the study area. Therefore, health education and behaviour change communication related to family planning services should be strengthened and access to family planning services should be improved.

## Introduction

Unmet need in family planning is defined as the percentage of women who don't want to be pregnant but are not using contraception. It indicates the gap between childbearing desire and contraceptive use, and calculated as the percentage of women of reproductive age, either married or in union, who have an unmet need for family planning [1].

Globally the prevalence of contraceptive use is increasing, however the unmet need for contraception remains a problem [2,3]. In 2019, 190 million (10%) of married women are estimated to have an unmet need for family planning. The prevalence of unmet need was higher in Africa (22%) and Oceania region (15%) compared to other parts of the world. Unmet need for contraceptives among married women was estimated at or below 10% [3,4].

Family planning services have a substantial impact on maternal morbidity and mortality. A rise in contraceptive prevalence rate and the decline in unmet need for family planning significantly reduces maternal mortality and disability associated with complications of pregnancy and child birth [5–11]. Despite this fact, unacceptably millions of women suffer from unintended pregnancies (both mistimed and unwanted) and related consequences worldwide every year [9–12].

Ethiopia is one of the twenty-two countries with a high maternal mortality rate worldwide in 2017 with maternal mortality rate ranging between 300 and 999 per 1000 live births. The maternal mortality rate in 2017 was 401 per 1000 live births [13]. The prevalence of unintended pregnancy in Ethiopia was 28% and Oromia region (33.8%) accounted the highest prevalence followed by Southern Nations, Nationalities, and People's region (30.6%). Satisfying demands for family planning leads to fewer unintended pregnancies, abortions, and child and maternal deaths [14].

The current level of unmet need for family planning was unacceptability high in Ethiopia standing at 22% in 2020 that is far away from the target set by the country (10%) [15]. Unmet needs for family planning varies with socio-demographic factors, residence, and geographic regions, with the highest unmet needs being in the Oromia region accounting to 29%. Unmet need for family planning had declined steadily in the period between 2000 to 2016 [16] and the prevalence of contraceptives had increased in the period from 2000 (8%) to 2019 (44%) [17].

Proper understanding of the extent of an unmet need for family planning and the factors associated with it has paramount importance in addressing the problems related to non-use of family planning services. It has substantial contributions to improve the health conditions of women and children and the underlying socio-economic problems of the country. There were studies that investigated the unmet need for family planning and associated factors in Ethiopia [18–25], and little evidence exists regarding unmet need for family planning particularly in

settings where there was growing industrialization and increased internal migration. Rapid urbanization had profound effect on population health [26]. There may be contextual differences among the study participants in different settings. Therefore, this study aimed to determine the unmet need for family planning services and identify factors associated with it among currently married women in Bishoftu town, Eastern Ethiopia. The study was focused in urban setting with a growing industrialization and aimed to identify socio-demographic and economic factors, reproductive health factors, and service characteristics that predict an unmet need for family planning among currently married women.

## Methods and materials

### Study setting and period

The study was conducted in Bishoftu town. Bishoftu town is one among the town administrations of Oromia region located 47 KM far away from Addis Ababa in Eastern part of Ethiopia. It is located adjacent to Eastern industrial zone of Ethiopia. The town administration comprises of 14 kebeles of which nine were urban kebeles and the remaining three were semi-urban kebeles. In the town, there were two public hospitals, five health centers, two private hospitals, and ten private clinics providing services ranging from preventive and basic curative to advanced medical services to the catchment area population. According to Bishoftu town health office report of 2020, the town had a total population of 217,971 and of this female population accounted 51% (111,165) and women in the reproductive age group eligible to family planning services were 19% (41,328). The study was conducted from 1st January to 28th February, 2021.

### Study design and participants' selection

A community-based cross-sectional study was conducted among currently married women of reproductive age group. All currently married women in the reproductive age group (15–49) and lived in the town for at least 6 months were eligible to the study. Infecund women, women who were not legally married and who were critically ill during the survey period were excluded. Infecundity and marital status of women were identified based on self-report.

The sample size was determined using Epi Info sample size calculator for cross-sectional surveys considering the assumptions and parameters: 95% confidence level, 4% margin of error, proportion of unmet need for family planning as 30.9% from the study conducted in Debre Birhan [27], 1.5 design effect, and 10% non-response rate. The calculated sample size yields 846.

$$\text{Sample size (n)} = \frac{(Z^2 * P * Q)}{d^2}$$

Where:

■ Confidence level (Z) = 1.96

■ Prevalence of outcome in the population (P) = 30.9%

■ Q = 1-P = 69.1%

■ Margin of error ($d^2$) = 4%

Multi-stage stratified sampling strategy was used to select kebeles and study participants. In the first stage, five urban kebeles and three semi-urban kebeles were selected using simple random sampling strategy from nine urban and five semi-urban kebeles. In the second stage, a

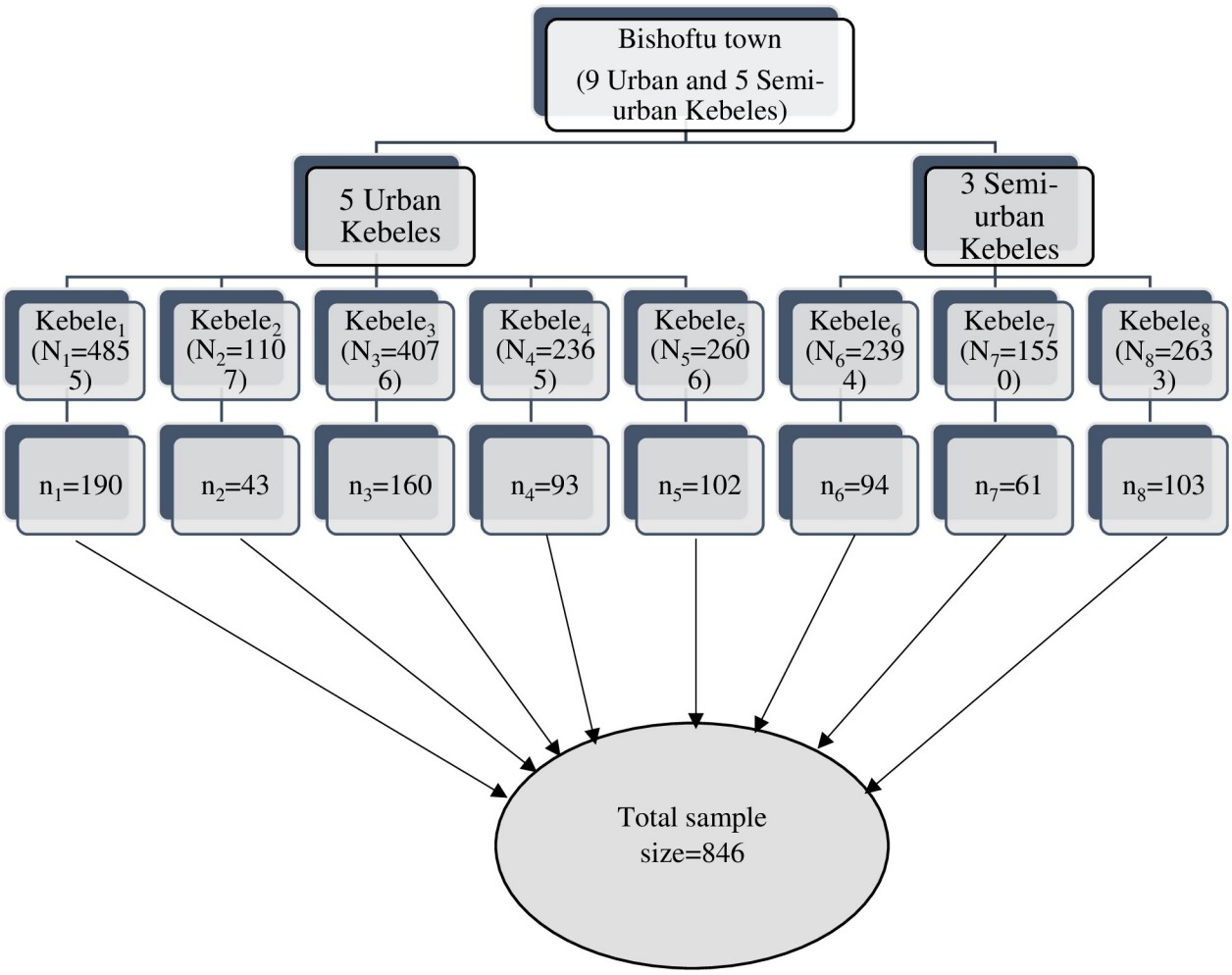

**Fig 1. Sample size and sampling strategy.**

systematic random sampling strategy was used to select households in each kebeles. All households in the primarily selected Kebeles and women in the reproductive age group were listed. The sample size was proportionally allocated to each selected Kebeles based on the total number of currently married women in each Kebeles. All eligible women in every 25th household (K = 25) were interviewed after taking consent for participation. In the case when the selected households had no eligible women, the next household was considered and whenever there were more than one eligible women in the sampled household, one woman was selected randomly. Each selected household was visited three times on occasions where respondents were unavailable during the first visit and after the third visit households were recorded as no-response (Fig 1).

## Study variables

The outcome variable was the unmet need for family planning. It was the sum of the unmet need for spacing and the unmet need for limiting. Other outcome variables included demand for family planning and demand satisfied for family planning. Demand for family planning was calculated as the sum of currently married women who were on family planning and the

unmet need for family planning. Percentage of demand satisfied for family planning was calculated as currently using family planning divided by the demand for family planning. The explanatory variables were socio-demographic and economic factors, reproductive health factors, and service characteristics.

## Data collection tools and procedures

Semi structured interviewer administered questionnaire was used to collect data. The questionnaire was developed based on relevant literature and adapted to the research context [21,23,27,28]. It comprised of three parts: background of respondents, reproductive characteristics, and service related factors. The questionnaire was primarily developed in English language and translated to local languages (Afan Oromo and Amharic).

Five nurses and two supervisors with qualification of bachelor of science in public health were participated in the data collection process. A two-day training was given to data collectors and supervisors on the questionnaire, data collection process and research ethics. Pre-test was conducted in 5% of the sample size in an adjacent Kebele of the study area and corrections were made to the questionnaire as appropriate. Supervisors have closely monitored the data collection process and provided support at the field level.

## Data processing and analysis

Each data records were checked for completeness and consistency, and duplicated cases were removed. Data were entered to Epi-Info version 7 and exported to SPSS version 25 for analysis. The data analysis was progressed in such a way that primarily descriptive statistics was used to describe and summarize the characteristics of respondents. Secondly, bivariate logistic regression was undertaken and those variables with P-value<0.25 were taken to multivariate logistic regression. The outcome variable of the study was the unmet need for family planning services. Variable Inflation Factor (VIF) was used to check the presence of multi collinearity and Hosmer-Lemeshow test of Goodness-of-fit was used to test how well the model explains the data. The strength of association was expressed in an odds ratio with 95% confidence interval and P-value <0.05 was used as cut-off point to declare significance in the final model.

## Ethical consideration

Research ethical clearance was obtained from Adama Hospital Medical College Institutional Research Ethics Review Board (Reference number: AHMC/MPHWek/8/12/2020). Support letter was taken from Oromia Regional Health Bureau and Bishoftu town health office. The research was conducted according to the Declaration of Helsinki. The research aims, benefits, and risks were explained to each research participant. Following this, a written informed consent was obtained from participants and for minors, informed written consent was taken from parents or guardians. No personal identifiers were recorded and codes were used on each questionnaire. Paper based data was kept in a locked cabinet and computer-based data were secured with a confidential password. Research data will only be used for the intended aim and not shared with the third people.

## Results

Eight hundred twenty-eight currently married women in the reproductive age group were participated in the study with a response rate of 97.8%.

## Sociodemographic and economic characteristics of women

The mean age of women was 31 years (SD 6.6) with a minimum age of 17 years and maximum age of 49 years. Four in ten (42%) of women were in the age group of 26–35 followed by women aged less than 25 years. Higher proportion of women were Orthodox Christians (36%) and had higher educational status (38%) (Table 1).

## Reproductive health characteristics

Age at first marriage was as early as 17 years and as late as 35 years with a mean age of 22 years (SD 3.3). Mean age at first pregnancy was 18 years. Thirty-eight percent of women had a desire to have two and fewer children and higher than half (54%) of them had 1–2 living children.

**Table 1.  Socio-demographic and economic characteristics of women in Bishoftu town, Eastern Ethiopia, January 2021.**

| Variable | Frequency (N = 828) | Percent |
|---|---|---|
| Age in years | | |
| ≤25 | 224 | 27 |
| 26–35 | 350 | 42 |
| 36–45 | 174 | 21 |
| ≥46 | 80 | 10 |
| Ethnicity | | |
| Oromo | 395 | 48 |
| Amhara | 217 | 26 |
| Tigre | 85 | 10 |
| Somali | 69 | 8 |
| Guraghe | 56 | 7 |
| Others* | 6 | 1 |
| Religion | | |
| Orthodox | 297 | 36 |
| Muslim | 129 | 16 |
| Protestant | 176 | 21 |
| Wakefata | 122 | 15 |
| Catholic | 104 | 13 |
| Educational status | | |
| No education | 166 | 20 |
| Primary | 91 | 11 |
| Secondary | 257 | 31 |
| Tertiary | 314 | 38 |
| Occupational Status | | |
| Government employee | 109 | 13 |
| Self-employee | 280 | 34 |
| Merchant | 112 | 14 |
| House wife | 242 | 29 |
| Daily labourer | 85 | 10 |
| Monthly income in Ethiopian Birr(ETB) | | |
| ≤2500 | 254 | 31 |
| 2500–5000 | 299 | 36 |
| 5001+ | 275 | 33 |

*Others: Kabata (n = 3), Wolayita (n = 3).

Twelve percent of women were currently pregnant with 8% of them wanting the pregnancy now, 2% wanted the pregnancy latter, and 2% not wanted the pregnancy at all (Table 2).

## Current contraceptive use patterns and service characteristics

Four hundred thirty-five (53%) women were using contraceptives at the time of the survey and out of which 107 (25%) and 328(75%) of the women were using contraceptives for limiting and spacing purposes, respectively. The most commonly used family planning method was implants 214(49%) followed by injectable 110(25%) and IUCD 84(19%). Pills accounted of 20 (5%) and condoms accounted of 7(2%).

Slightly greater than half (54%) of women were visited by health care providers in the last 12 months of the survey period, and the round trip to visit a health institutions was less than or equals to 30 minutes for most (98%) of the women surveyed. Four-hundred sixty (56%) of women spent less than 15 minutes in health institutions and greater than half (53%) of women told about family planning options in their current visits. Appointment was given to 52% of women (Table 3).

## Unmet need for family planning

Unmet need for family planning was calculated by summing up the number of women who do not use family planning because of fear of side effects, not having sex, infrequent sex, and health concerns, but they want the child later and the other did not want at all.

**Table 2. Reproductive health characteristics of currently married women of reproductive age in Bishoftu Town, Eastern Ethiopia, January 2021.**

| Variable | Frequency (N = 828) | Percent (95% CI) |
|---|---|---|
| Age at first marriage in years | | |
| ≤19 | 262 | 32(28.5–34.9) |
| 20–24 | 316 | 38(34.9–41.5) |
| ≥25 | 250 | 30(27.1–33.4) |
| Age at first pregnancy in years (N = 719) | | |
| ≤ 19 | 52 | 7(5.5–9.3) |
| 20–24 | 449 | 62(58.9–65.9) |
| ≥25 | 218 | 30(27.0–33.8) |
| Desired number of children | | |
| ≤ 2 | 313 | 38(34.5–41.2) |
| 3–4 | 259 | 31(28.2–34.5) |
| 5+ | 256 | 31(27.8–34.1) |
| Number of living children | | |
| No child | 133 | 16(13.7–18.7) |
| 1–2 | 444 | 54(50.2–57.0) |
| 3–4 | 236 | 29(25.5–31.7) |
| 5+ | 15 | 2(1.1–2.9) |
| Currently pregnant | | |
| Yes | 96 | 12(9.6–13.9) |
| No | 732 | 88(86.1–90.5) |
| The current pregnancy was (N = 96) | | |
| Wanted now | 63 | 66(55.7–74.6) |
| Wanted latter | 19 | 20(12.7–28.7) |
| Not wanted at all | 14 | 15(8.6–22.7) |

Abbreviations: *CI* confidence interval.

**Table 3. Family planning service characteristics among currently married women of reproductive age in Bishoftu Town, Eastern Ethiopia, January 2021.**

| Variable | Frequency | Percent (95% CI) |
|---|---|---|
| Visited by Family planning provider in the last 12 months | | |
| Yes | 444 | 54(50.2–57.0) |
| No | 384 | 46(43.0–49.8) |
| Time taken for round trip for Family planning service | | |
| ≤ 15 minutes | 402 | 49(45.2–52.0) |
| 16–30 minutes | 410 | 49(46.1–52.9) |
| 31–60 minutes | 16 | 2(1.2–3.1) |
| Time taken in the health facility | | |
| ≤15 minutes | 460 | 56(52.2–58.9) |
| 16–30 minutes | 256 | 31(27.8–34.1) |
| 31–60 minutes | 112 | 14(11.3–16.0) |
| Discussed about family planning options | | |
| Yes | 442 | 53(50.0–56.8) |
| No | 386 | 47(43.2–50.0) |
| Got family planning method wanted | | |
| Yes | 516 | 62(59.0–65.6) |
| No | 312 | 38(34.4–41.0) |
| Provider give appointment in the current visit | | |
| Yes | 433 | 52(49.0–55.7) |
| No | 395 | 48(44.3–51.1) |

Abbreviations: *CI* confidence interval.

Unmet need for family planning among currently married women in reproductive age groups was 217 [26%, 95% CI: 23–29]. Out of this, 129 (16%) were unmet need for spacing and 88(11%) were unmet need for limiting. Demand for family planning in the study area was the sum of currently on family planning 435(53%) and current unmet need for family planning 217(26%), which was 79%. Percentage of demand satisfied for family planning was 67% (that is currently on family planning (n = 435) divided by demand for family planning (n = 652) (Fig 2).

The major reason of not using the family planning method was wanting more children 150 (38%) followed by not having sex due to their husbands were on field work 102(26%). Fear of side effects accounted of 59(15%), infrequent sex accounted of 50(13%), health concern accounted of 24(6%), and menopause/ hysterectomy accounted of 12(3%).

## Factors associated to the unmet need for family planning

In the bivariate logistic regression analysis, maternal age, educational status, occupation status, total birth, desired number of children, income, age at first pregnancy, number of pregnancies, being visited by a health care provider in the last 12 months, main place where they use family planning, and age at first marriage were identified as candidate variables with P-value<0.25.

In the multivariate logistic regression analysis, only variables including maternal age, educational status, occupational status, desired number of children, and being visited by a health care provider in the last 12 months were statistically associated with an unmet need for family planning after controlling other confounders.

The interpretations were given as below. The odds of unmet need for family planning was three times more likely among women with age less than 25 years compared to women with

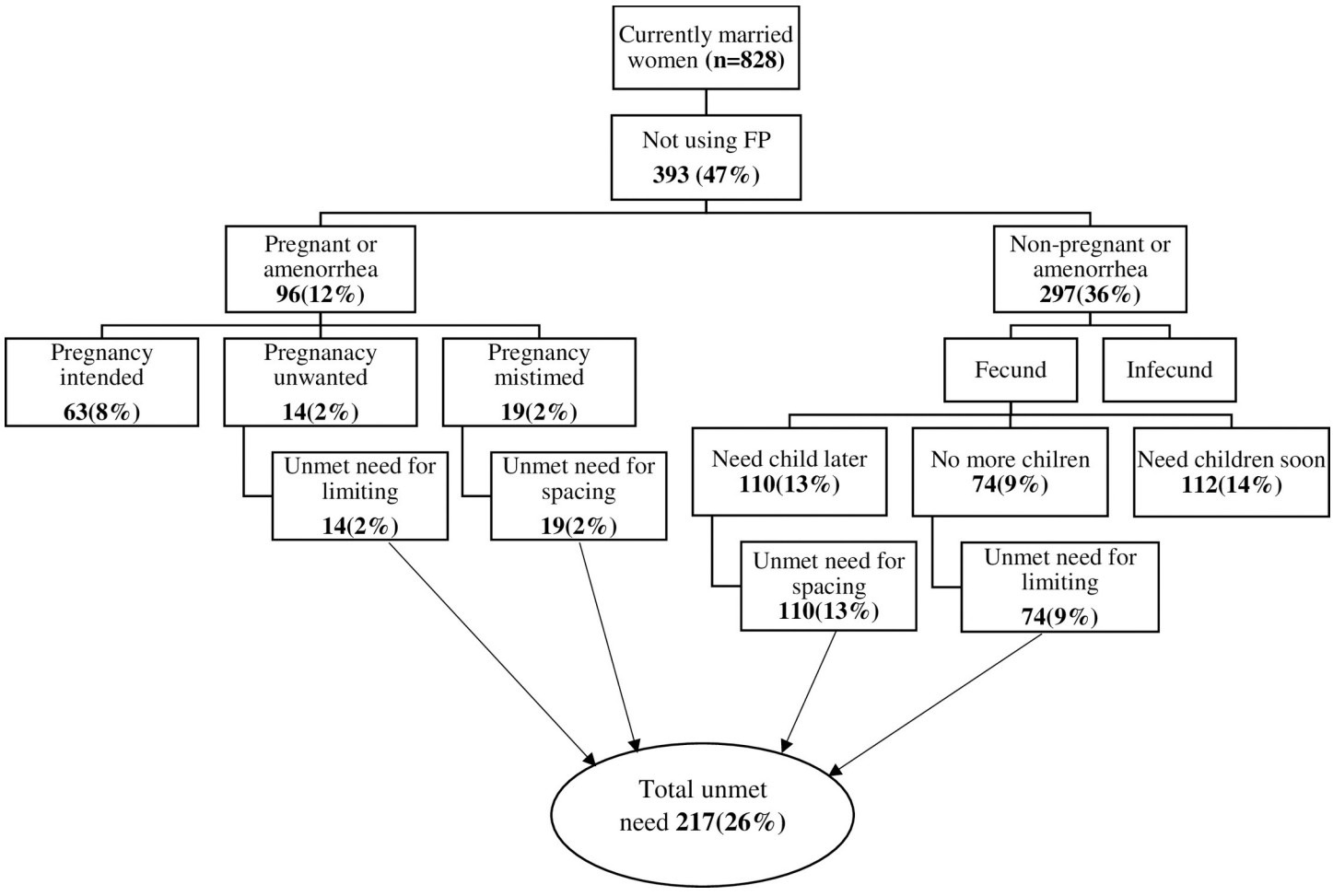

**Fig 2. Schematic diagram of the unmet need for family planning among currently married women.**

age greater than 46 years [AOR = 3.0, 95% CI:1.51–5.95]. The odds of unmet need for family planning was 2.5 times more likely among women with primary education compared to women with tertiary education [AOR, 2.49, 95% CI: 1.22–5.07]. The odds of unmet need for family planning was two times more likely among self-employed women [AOR, 1.98, 95% CI: 1.15–3.39] and house wife's [AOR, 1.78, 95% CI: 1.02–3.12] compared to daily labourers, respectively. Women who had not been visited by health care providers within the last 12 months prior to the survey date were two times more likely to have an unmet need for family planning when compared to women who had been visited [AOR, 1.81, 95% CI: 1.26–2.60]. The odds of unmet need for family planning was 1.53 times more likely among women who wanted to give birth of ≤2 children compared to women who wanted five and more children [AOR, 1.53, 95% CI: 1.01–2.30] (Table 4).

## Discussion

The study showed that a quarter of currently married women in the reproductive age group (26%, 95% CI: 23,29) had an unmet need for family planning in the study area. Sixteen percent of women had an unmet need for spacing and one in ten of women had an unmet need for limiting. Women age, occupational status, educational status, desired number of children, and

**Table 4. Factors associated with the unmet need for family planning.**

| Variables | Unmet need for FP | | COR (95% CI) | AOR (95% CI) |
|---|---|---|---|---|
| | Yes (%) | No (%) | | |
| Maternal age in years | | | | |
| ≤ 25 | 24(2.9) | 200(24.2) | 1.20[1.00–3.49] | 3.00 [1.51–5.95] * |
| 26–35 | 114(13.8) | 236(28.5) | 2.23[1.63–3.46] | 0.83[0.47–1.47] |
| 36–45 | 59(7.1) | 115(13.9) | 0.9[(0.99–1.23] | 0.73[0.39–1.35] |
| ≥ 46 | 20(2.4) | 60(7.2) | 1.0 | 1.0 |
| Occupational status | | | | |
| Government employee | 33 (4) | 76(9.2) | 0.67[0.37–1.22] | 1.65[0.87–3.13] |
| Self-employee | 68 (8.2) | 212(25.6) | 1.49 [1.18–2.86] | 1.98[1.15–3.39]* |
| Merchant | 29 (3.5) | 83(10) | 0.54[0.29–0.99] | 1.18[0.61–2.29] |
| House wife | 54 (6.5) | 189(22.8) | 1.44 [1.26–2.75] | 1.78[1.02–3.12]* |
| Daily labourer | 33 (4) | 51 | 1.0 | 1.0 |
| Educational status | | | | |
| No education | 53 (6.4) | 113 (13.6) | 1.04 [0.58–1.86] | 0.78[0.49–1.21] |
| Primary education | 12 (1.4) | 79 (9.5) | 1.71 [1.0–2.91] | 2.49[1.22–5.07]* |
| Secondary education | 76 (9.2) | 181 (21.9) | 0.77 [0.53–1.13] | 0.68[0.45–1.03] |
| Collage and above | 76(9.2) | 238(28.7) | 1.0 | 1.0 |
| Being visited by health care provider in the last 12 months | | | | |
| No | 102(12.3) | 340(41.1) | 3.71[2.66–5.17] | 1.81[1.26–2.60]* |
| Yes | 115(13.9) | 271(32.7) | 1.0 | 1.0 |
| Number of children desired | | | | |
| ≤2 | 64(8.1) | 249(30.2) | 1.79[1.49–2.27] | 1.53[1.01–2.31]* |
| 3–4 | 81(9.4) | 178(21.4) | 0.21[0.13–0.35] | 1.14[0.76–1.72] |
| 5+ | 72(8.7) | 184(22.2) | 1.0 | 1.0 |

* Statistically significant at *P-value* < 0.05.

Abbreviations: *COR* crude odds ratio, *AOR* adjusted odds ratio, *CI* confidence interval, 1.0 reference category.

being visited by a health care provider in the last 12 months were statistically associated with an unmet need for family planning.

The prevalence of unmet need for family planning in the study area was higher than the national average (22%) and slightly lower than the regional average (29%) [16]. It is far from the target set at national level that is 10% in the year 2020 [15]. The prevalence was higher compared with the United Nations sphere standard of unmet need for family planning, which is considered to be high if greater than 25% and the global estimate of unmet need for family planning among women in the reproductive age group (24.3%) [3,19]. The finding was also higher than studies conducted in different parts of the world including Zambia (21%) [29], Myanmar (19.4%) [30] and Bangladesh (13.5%), [31] and Kutaye district, Western Ethiopia (23.1%) [22]. Whereas, the finding was comparable to cross-sectional studies conducted in Tiro Afata (26%) [23], Misha (26.5%) [32], and Damot Woyde Woredas (26.3%) [20].The variations in the unmet need for family planning might be related to contextual variations including health services coverage, knowledge and attitudes towards family planning services, and socio-demographic and economic factors. On the contrary, studies conducted in Hargeisa, Somaliland (30%) [33], India (55.3%) [34], and Asebot town (37%) [21] and Debre Birhan town (31%) [27] in Ethiopia have shown a higher unmet need prevalence compared to the study area. Likewise, a systematic review and meta-analysis on studies done in different parts of Ethiopia showed slightly higher rate of unmet need for family planning among married

women (32.8%) [25] and among women in the reproductive age group that ranged from 26.5% to 36.4% [19]. The higher prevalence of unmet need for family planning might be because of low access to contraceptives, low awareness towards modern contraceptive methods, and low partner involvement in such matters [18,35].

Demand for family planning was satisfied for two-thirds of currently married women. This was slightly higher than the national average (61.6%) and higher than Oromia region (48.9%) [16]. In the study area, only 53% of currently married women were using contraceptives and the majority of them (75%) use contraceptives for spacing purposes. The most commonly used methods of contraceptives were implants, injectable and IUCD, accounting of 93%. This was consistent to the national survey [36]. If all currently married women who said that they wanted to space and limit their children were to use family planning methods, the contraceptive prevalence in the study area would increase from 53% to 79%. The main reasons for non-use of family planning among currently married women were wanting more children, not having sex, fear of side effects, infrequent sex, health concern, and menopause, which is a result comparable with other studies else [24,37]. Thus, providing a comprehensive information of family planning methods enhances clients to use the most suitable methods and helps to attain SDG 3.7 of informed choice of contraceptive methods [38].

Unmet need for family planning has consequences on women and their families such as unintended pregnancy that in turn leads to increased maternal morbidity and mortality. It also leads to physical abuse and affects the overall wellbeing of women. Hence, addressing the family planning needs of women is critical in the effort towards attaining a goal related to maternal mortality [38,39]. In Ethiopia in the past two decades, the prevalence of contraceptives among currently married women has increased, although not as intended and stood at 41% in 2019 [17]. Unmet need for family planning had declined steadily from 36% in 2000 to 22% in 2016 [16]. This implies the need to accelerate efforts towards Sustainable Development Goals related to family planning.

Our study showed that factors such as socio-demographic, economic, reproductive health, and being visited by a health care providers were significantly associated with the unmet need for family planning. The odds of unmet need for family planning were more likely in the younger age group of currently married women than the older age group women. This finding was comparable to other parts of the world whereas in contrast to studies done in different parts of Ethiopia [21,28,29,31,40,41]. This might be because older women are mature and better decide on their health including the use of family planning services compared to younger women. Besides, older woman had less desire to children compared to younger women, which increases the need to use family planning methods. As the consequence of this a younger age groups of women should be targeted for family planning services in the study area.

The study also indicated the relationship between maternal education and occupational status with an unmet need for family planning. Odds of unmet need for family planning were more likely in women with primary education than college and above. This finding was comparable to a study conducted in different parts of the World and in Ethiopia [20,22,29,33,40,42]. Moreover, the odds of unmet need were more likely among self-employed women and housewife's compared to daily labourers. This might be because self-employed women and those who were housewives are relatively stable and better decide on their preferences compared to daily labourers, which usually struggles to maintain their daily needs. Daily labourers are more likely to have control on their life situations with the number of children and accept and implement health actions related to family planning better than their counterparts. This finding was comparable to a study conducted in different parts of Ethiopia [25,27].

In the study area, women who were visited by a health care provider in the last 12 months of the survey date more likely had met the need for family planning services compared to

women who were not visited. This is consistent with a study done in other settings of Ethiopia [22,23,27,41]. One of the service delivery modalities in the urban health extension program of Ethiopia is home to home or outreach services. Using this modality, the health extension professional provides health education and behaviour change communication related to family planning and other packages of services. They also provide family planning services to women who are less accessible to health facilities. The increased awareness and knowledge of women towards family planning services enable them to have access to services and hence, met needs [43–45].

Desire to have children was the other factor that affects the unmet need for family planning among currently married women. The odds of an unmet need for family planning was more likely among women who desired to have fewer children (≤ 2 children) than those who desired to have more children (more than 5 children). As the number of children increases, the probability of using family planning services will increase. This finding is consistent with other studies [20–23,29,33,40].

The study has included a larger sample sizes and this enables appropriate generalisation of findings. The study was focused only on currently married women and doesn't able to capture the important factors that determine the unmet need for family planning in other groups such as women who are in union and not married.

## Conclusions

Though modern contraceptive is a highly cost-effective public intervention in reducing maternal mortality [8,46,47], the current study indicated that the unmet need for family planning among currently married women was higher compared to the national average and slightly lower than the regional average. This has considerable implications on women and the society at large, including unintended pregnancy, unsafe abortions, physical abuse, and overall wellbeing. It increases maternal morbidity and mortality, and affects efforts toward improving the health status of women and the economic development of the nation.

Unmet need for family planning was associated with socio-demographic factors (maternal age), economic factors (educational and occupational status), desire to have children, and access to a health care provider. Hence, an effort should be made to strengthen access to family planning methods and health care providers in the study area. Health education and behaviour change communication strategies should be accessible to target women in the study area. This can be done by using health extension professionals and strengthening the urban health extension program, which has been under implementation since 2009 in Ethiopia. It also requires strengthening the women development army at the grass root level that helps to identify barriers to service utilization and enable access to health services through providing health information.

## Supporting information

**S1 File.**
(ZIP)

**S1 Dataset.**
(XLSX)

## Acknowledgments

Our acknowledgment should go to the different levels of administrative hierarchy and study participants.

## Author Contributions

**Conceptualization:** Megersa Girma Garo, Sileshi Garoma Abe, Worku Dugasa Girsha, Dawit Wolde Daka.

**Data curation:** Megersa Girma Garo, Sileshi Garoma Abe, Worku Dugasa Girsha, Dawit Wolde Daka.

**Formal analysis:** Megersa Girma Garo, Sileshi Garoma Abe, Worku Dugasa Girsha, Dawit Wolde Daka.

**Funding acquisition:** Megersa Girma Garo, Sileshi Garoma Abe, Worku Dugasa Girsha, Dawit Wolde Daka.

**Investigation:** Megersa Girma Garo, Sileshi Garoma Abe, Worku Dugasa Girsha, Dawit Wolde Daka.

**Methodology:** Megersa Girma Garo, Sileshi Garoma Abe, Worku Dugasa Girsha, Dawit Wolde Daka.

**Project administration:** Megersa Girma Garo, Sileshi Garoma Abe, Worku Dugasa Girsha, Dawit Wolde Daka.

**Resources:** Megersa Girma Garo, Sileshi Garoma Abe, Worku Dugasa Girsha.

**Software:** Sileshi Garoma Abe, Worku Dugasa Girsha, Dawit Wolde Daka.

**Supervision:** Megersa Girma Garo, Sileshi Garoma Abe, Worku Dugasa Girsha, Dawit Wolde Daka.

**Validation:** Megersa Girma Garo, Sileshi Garoma Abe, Worku Dugasa Girsha, Dawit Wolde Daka.

**Visualization:** Dawit Wolde Daka.

**Writing – original draft:** Dawit Wolde Daka.

**Writing – review & editing:** Dawit Wolde Daka.

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
