## [Decision Letter · Decision Letter 0]

16 Sep 2021

PONE-D-21-23218Unmet need for family planning and associated factors among currently married women of reproductive age in Bishoftu town, Eastern Ethiopia.PLOS ONE

Dear Dr. Daka,

Thank you for submitting your manuscript to PLOS ONE. After careful consideration, we feel that it has merit but does not fully meet PLOS ONE’s publication criteria as it currently stands. Therefore, we invite you to submit a revised version of the manuscript that addresses the points raised during the review process.

ACADEMIC EDITOR: Considering the reviewers suggestion and my own reading, I am suggesting a minor revision to this paper. Can you develop the discussion a little more so that it will be relevant to global readers. Compare your results with other developing countries. For instance, use following references to make a comparative discussion. Rana MJ, Goli S, Mishra R, Gautam A, Datta N, Nanda P, Verma R. Contraceptive Method Information and Method Switching in India. Sustainability. 2021; 13(17):9831. https://www.mdpi.com/2071-1050/13/17/9831 Misra S, Goli S, Rana MJ, Gautam A, Datta N, Nanda P, Verma R. Family Welfare Expenditure, Contraceptive Use, Sources and Method-Mix in India. Sustainability. 2021; 13(17):9562. https://doi.org/10.3390/su13179562 Rana, M. J., & Goli, S. (2021). The road from ICPD to SDGs: health returns of reducing the unmet need for family planning in India. Midwifery, 103107. The road from ICPD to SDGs: Health returns of reducing the unmet need for family planning in India - ScienceDirect Goli, S., Moradhvaj, James, K.S., Singh, D. & Srinivasan, V. (2020). Road to family planning and RMNCHN related SDGs: Tracing the role of public health spending in India, Global Public Health, DOI: 10.1080/17441692.2020.1809692. Goli S., Gautam A, Rana MJ, Ram H, Ganguly D, Reja T, Nanda P, Datta N, Verma R. (2020). Is unintended birth associated with physical intimate partner violence? Evidence from India. Journal of biosocial science.1-16.

We look forward to receiving your revised manuscript.

Kind regards,

Srinivas Goli, Ph.D.

Academic Editor

PLOS ONE

Additional Editor Comments (if provided):

Considering the reviewers suggestion and my own reading, I am suggesting a minor revision to this paper. Can you develop the discussion a little more so that it will be relevant to global readers. Compare your results with other developing countries. For instance, use following references to make a comparative discussion.

Rana MJ, Goli S, Mishra R, Gautam A, Datta N, Nanda P, Verma R. Contraceptive Method Information and Method Switching in India. Sustainability. 2021; 13(17):9831. https://www.mdpi.com/2071-1050/13/17/9831

Misra S, Goli S, Rana MJ, Gautam A, Datta N, Nanda P, Verma R. Family Welfare Expenditure, Contraceptive Use, Sources and Method-Mix in India. Sustainability. 2021; 13(17):9562. https://doi.org/10.3390/su13179562

Rana, M. J., & Goli, S. (2021). The road from ICPD to SDGs: health returns of reducing the unmet need for family planning in India. Midwifery, 103107. The road from ICPD to SDGs: Health returns of reducing the unmet need for family planning in India - ScienceDirect

Goli, S., Moradhvaj, James, K.S., Singh, D. & Srinivasan, V. (2020). Road to family planning and RMNCHN related SDGs: Tracing the role of public health spending in India, Global Public Health, DOI: 10.1080/17441692.2020.1809692.

Goli S., Gautam A, Rana MJ, Ram H, Ganguly D, Reja T, Nanda P, Datta N, Verma R. (2020). Is unintended birth associated with physical intimate partner violence? Evidence from India. Journal of biosocial science.1-16.

3. You indicated that you had ethical approval for your study. In your Methods section, please ensure you have also stated whether you obtained consent from parents or guardians of the minors included in the study or whether the research ethics committee or IRB specifically waived the need for their consent.

Reviewers' comments:

Reviewer's Responses to Questions

**Comments to the Author**

1. Is the manuscript technically sound, and do the data support the conclusions?

Reviewer #1: Yes

Reviewer #2: No

2. Has the statistical analysis been performed appropriately and rigorously? 

Reviewer #1: Yes

Reviewer #2: No

3. Have the authors made all data underlying the findings in their manuscript fully available?

Reviewer #1: No

Reviewer #2: Yes

4. Is the manuscript presented in an intelligible fashion and written in standard English?

Reviewer #1: Yes

Reviewer #2: No

5. Review Comments to the Author

Reviewer #1: The title of the paper should modified first.

Give some more information on study design and participant selection. Literally the selection process sounds good, but authors must put some statistical formula for sample size estimation.

Firstly the things mentioned in data processing and analysis were not truely done.

Except table four rest tables were depicting only number and percentages.

Bivariate logistic regression and multivariate logistic regression analysis tables were not present in the manuscript seperately.

Dicussion should be more concentrated on the regression tables, as from table four it is clear that last category of each variable is reference category, so author must discuss his/her results based on the reference category, whether in respect of the reference catgory other catgories are increasing or decreasing.

Reviewer #2: n the abstract, authors should consider a comma before including. Contrary to the statement that little is known about determinants of unmet need for family planning in the study area, much research study has been done. Discard the repetition of Ethiopia. Change primary outcome to response variable or outcome variable. The adjusted odds ratio should be in decimal (AOR=3), it should be 3.0. Authors should check power of the test, looking at the confidence interval, p-value and effect size.

Introduction session: Grammatical blunders frequent often. In the conceptualisation of unmet need, the statement "women either who wishes.... The key problems associated with the issue of unmet need for family planning were not discussed. Apart from that, the argument that much research has not been done is not convincing and evidence-based. Many research have been done on determinants of unmet need for family planning in this context. As such, authors need to argue clearly and make the problem statement and research gap convincing. Also, the authors should provide a clear context for the research. There should be clear justification for researching married women in the urban context. Unmet need for family planning varies significantly..... (references) missing. The authors have a weak academic writing style and sloppy statement. The authors wrote there were studies without reference in Ethiopian context? Thus, why this study? The authors should state the current state of evidence in the study area and argue clearly on what is not known. Check grammar eligible for or to?Line 115, each participant household was visited.

Recorded as no response should be correct. Consider: socio-demographic and socio-economic?

Method session

The justification for the study area was not clearly stated. Data were checked for completeness The data analysis was progressed in such a way that primarily descriptive statistics to describe and ....check the sentence is clumsy and sloppy. There is an un-academic statement: candidate variable. What informed the choice of p<0.25 consideration for multivariate analysis. Why logistic regression? Is it binary, ordinal or multinomial?

Results

The descriptive statistics and interpretation should be reduced. Authors should check the presentation of results at the three levels of analysis. The results should follow the objectives of the study.

Discussion: This section was poorly written. The introductory paragraph should focus on the overall aim of the study and what was found. Each of the objectives should be considered in each paragraph. The authors should present results, comparison with previous studies, plausible explanation for divergence and convergence of results. The authors should consider date of publication, study population, settings, sample size in providing possibles reasons for deviation of results.

There are many counter-intuitive results. Thus, there is a need to explore the reasons for the deviation. Authors should consider the statement: the odds of having unmet need for family planning was more likely among women who desire to have fewer children.

Does not should be written in full in formal writing, such as this.

The study has no policy implications, discussion on social reality of effect size and other confounders.

Conclusion

There was no clear philosophic stance in the conclusion. Many blanket statements were made in a sweeping manner.

APA format of referencing should be followed.

6. PLOS authors have the option to publish the peer review history of their article (what does this mean?). If published, this will include your full peer review and any attached files.

Reviewer #1: **Yes: **Tushar Dakua

Reviewer #2: **Yes: **Olufemi M. Adetutu, PhD

---

## [Author Response · Author response to Decision Letter 0]

31 Oct 2021

31th October, 2021

To Editor

PLOS ONE

Subject: Submitting revised manuscript 

It is our great pleasure for the opportunity given to us to submit the revised version of the manuscript and we acknowledge the fruitful comments provided by the reviewers. Our point-by-point responses to each of the reviewers’ comments are presented as follows.

Reviewer 1:

1. The title of the paper should have modified first. Give some more information on study design and participant selection. Literally the selection process sounds good, but authors must put some statistical formula for sample size estimation.

Response: Thank you very much. We appreciate the comment given by reviewer to modify the title. We haven’t modified the title of the paper because we thought that the current title is appropriate for the contents of the paper. We have included sample size calculation formula and provide information about study design and participant selection. 

2. Firstly the things mentioned in data processing and analysis were not truly done.

Except table four rest tables were depicting only number and percentages. Bivariate logistic regression and multivariate logistic regression analysis tables were not present in the manuscript separately.

Response: Thank you very much. We have included 95% CI for percent for table 2 and 3. We have presented both the bivariate and multivariate regression tables because the number of tables might increase beyond the journals limit. 

3. Discussion should be more concentrated on the regression tables, as from table four it is clear that last category of each variable is reference category, so author must discuss his/her results based on the reference category, whether in respect of the reference category other categories are increasing or decreasing.

Response: Thank you very much for the important comment. We have discussed the most important findings of the study including the results of the logistic regression beginning from Line # 316. 

Reviewer 2:

1. In the abstract, authors should consider a comma before including. Contrary to the statement that little is known about determinants of unmet need for family planning in the study area, much research study has been done. Discard the repetition of Ethiopia. Change primary outcome to response variable or outcome variable. The adjusted odds ratio should be in decimal (AOR=3), it should be 3.0. Authors should check power of the test, looking at the confidence interval, p-value and effect size.

Response: Thank you very much. We have addressed it.

2. Introduction session: Grammatical blunders frequent often. In the conceptualization of unmet need, the statement "women either who wishes.... The key problems associated with the issue of unmet need for family planning were not discussed. Apart from that, the argument that much research has not been done is not convincing and evidence-based. Many research has been done on determinants of unmet need for family planning in this context. As such, authors need to argue clearly and make the problem statement and research gap convincing. Also, the authors should provide a clear context for the research. There should be clear justification for researching married women in the urban context. Unmet need for family planning varies significantly..... (references) missing. The authors have a weak academic writing style and sloppy statement. The authors wrote there were studies without reference in Ethiopian context? Thus, why this study? The authors should state the current state of evidence in the study area and argue clearly on what is not known. Check grammar eligible for or to? Line 115, each participant household was visited. Recorded as no response should be correct. Consider: socio-demographic and socio-economic?

Response: Thank you very much for the constructive comments. We have rewritten the problem statement or research gap statement as presented in the last paragraph of the introduction. We have also addressed grammar in sentence construction. 

3. Method session: The justification for the study area was not clearly stated. Data were checked for completeness The data analysis was progressed in such a way that primarily descriptive statistics to describe and ....check the sentence is clumsy and sloppy. There is an un-academic statement: candidate variable. What informed the choice of p<0.25 consideration for multivariate analysis. Why logistic regression? Is it binary, ordinal or multinomial?

Response: Thank you. We have corrected statements. We have used logistic regression because outcome variable is categorical. 

4. Results: The descriptive statistics and interpretation should be reduced. Authors should check the presentation of results at the three levels of analysis. The results should follow the objectives of the study.

Response: Thank you for the comment. We have presented the findings of the study based on the aim or research objective. Primarily, we have presented the characteristics of study participants followed by the outcome variable and associated factors. 

5. Discussion: This section was poorly written. The introductory paragraph should focus on the overall aim of the study and what was found. Each of the objectives should be considered in each paragraph. The authors should present results, comparison with previous studies, plausible explanation for divergence and convergence of results. The authors should consider date of publication, study population, settings, sample size in providing possible reasons for deviation of results. There are many counter-intuitive results. Thus, there is a need to explore the reasons for the deviation. Authors should consider the statement: the odds of having unmet need for family planning was more likely among women who desire to have fewer children. Does not should be written in full in formal writing, such as this. The study has no policy implications, discussion on social reality of effect size and other confounders.

Response: Thank you very much. Discussion is organized in such a way that in the first paragraph we have summarized the main findings of the study based on the objectives and in the successive paragraph we have discussed each of the main finings, compared findings with other related finings, explain discrepancies and tried to show implications of the findings. 

6. Conclusion: There was no clear philosophic stance in the conclusion. Many blanket statements were made in a sweeping manner. APA format of referencing should be followed.

Response: Thank you for the comment. We have revised conclusion statements. Because the journals requirement is Vancouver style of referencing, we have used it. 

Kind regards,

 Dawit Wolde Daka

Corresponding Author

E-mail: dave86520@gmail.com

31th October, 2021

To Editor

PLOS ONE

Subject: Submitting revised manuscript 

It is our great pleasure for the opportunity given to us to submit the revised version of the manuscript. A point-by-point response to the editors’ comments are presented as follows.

1. Can you develop the discussion a little more so that it will be relevant to global readers? Compare your results with other developing countries. For instance, use following references to make a comparative discussion.

Response: Thank you very much, we have addressed it. 

Response: Thank you very much, we have addressed it.

Response: Thank you. We have provided citations indicating list of references used in developing a questionnaire. Additionally, we have supplied the questionnaire as supporting information.

4. You indicated that you had ethical approval for your study. In your Methods section, please ensure you have also stated whether you obtained consent from parents or guardians of the minors included in the study or whether the research ethics committee or IRB specifically waived the need for their consent. 

Response: Thank you. Informed written consent was taken from parents or guardians for minors as indicated in the revised manuscript (Line # 172-173). 

5. In your Data Availability statement, you have not specified where the minimal data set underlying the results described in your manuscript can be found. PLOS defines a study's minimal data set as the underlying data used to reach the conclusions drawn in the manuscript and any additional data required to replicate the reported study findings in their entirety. All PLOS journals require that the minimal data set be made fully available. For more information about our data policy, please see http://journals.plos.org/plosone/s/data-availability . Upon re-submitting your revised manuscript, please upload your study’s minimal underlying data set as either Supporting Information files or to a stable, public repository and include the relevant URLs, DOIs, or accession numbers within your revised cover letter. For a list of acceptable repositories, please see http://journals.plos.org/ plosone/s/data-availability# loc-recommended-repositories. Any potentially identifying patient information must be fully anonymized.

Response: Thank you. We have supplied the minimal data set as supporting information file.

Response: Thank you. We have addressed it. 

Kind regards,

Dawit Wolde Daka

Corresponding Author

dave86520@gmail.com

---

## [Decision Letter · Decision Letter 1]

22 Nov 2021

Unmet need for family planning and associated factors among currently married women of reproductive age in Bishoftu town, Eastern Ethiopia.

PONE-D-21-23218R1

Dear Dr. Daka,

We’re pleased to inform you that your manuscript has been judged scientifically suitable for publication and will be formally accepted for publication once it meets all outstanding technical requirements.

Kind regards,

Srinivas Goli, Ph.D.

Academic Editor

PLOS ONE

Additional Editor Comments (optional):

Considering my own reading and reviewers opinion, I am recommending this paper for publication in PLOS One.

Reviewers' comments:

Reviewer's Responses to Questions

**Comments to the Author**

1. If the authors have adequately addressed your comments raised in a previous round of review and you feel that this manuscript is now acceptable for publication, you may indicate that here to bypass the “Comments to the Author” section, enter your conflict of interest statement in the “Confidential to Editor” section, and submit your "Accept" recommendation.

Reviewer #1: All comments have been addressed

2. Is the manuscript technically sound, and do the data support the conclusions?

Reviewer #1: Yes

3. Has the statistical analysis been performed appropriately and rigorously? 

Reviewer #1: Yes

4. Have the authors made all data underlying the findings in their manuscript fully available?

Reviewer #1: Yes

5. Is the manuscript presented in an intelligible fashion and written in standard English?

Reviewer #1: Yes

6. Review Comments to the Author

Reviewer #1: Earlier comments have been addressed. Authors have really work hard. I recommend editor to publish the paper.

7. PLOS authors have the option to publish the peer review history of their article (what does this mean?). If published, this will include your full peer review and any attached files.

Reviewer #1: **Yes: **Tushar Dakua

---

## [Editor Report · Acceptance letter]

25 Nov 2021

PONE-D-21-23218R1 

Unmet need for family planning and associated factors among currently married women of reproductive age in Bishoftu town, Eastern Ethiopia. 

Dear Dr. Daka:

I'm pleased to inform you that your manuscript has been deemed suitable for publication in PLOS ONE. Congratulations! Your manuscript is now with our production department. 

Kind regards, 

on behalf of

Dr. Srinivas Goli 

Academic Editor

PLOS ONE